# Neovascular Macular Degeneration: A Review of Etiology, Risk Factors, and Recent Advances in Research and Therapy

**DOI:** 10.3390/ijms22031170

**Published:** 2021-01-25

**Authors:** Arunbalaji Pugazhendhi, Margaret Hubbell, Pooja Jairam, Balamurali Ambati

**Affiliations:** 1Knights Campus for Accelerating Scientific Impact, University of Oregon, Eugene, OR 97403, USA; arunp@uoregon.edu (A.P.); mhubbell@uoregon.edu (M.H.); 2Vagelos College of Physicians & Surgeons, Columbia Irving Medical Center, Columbia University, New York, NY 10032, USA; poojajairam@yahoo.com

**Keywords:** age-related macular degeneration, neovascular age-related macular degeneration, etiopathogenesis, risk factors, gene-therapy, anti-VEGF, future advancements

## Abstract

Neovascular age-related macular degeneration (exudative or wet AMD) is a prevalent, progressive retinal degenerative macular disease that is characterized by neovascularization of the choroid, mainly affecting the elderly population causing gradual vision impairment. Risk factors such as age, race, genetics, iris color, smoking, drinking, BMI, and diet all play a part in nvAMD’s progression, with anti-vascular endothelial growth factor (anti-VEGF) therapy being the mainstay of treatment. Current therapeutic advancements slow the progression of the disease but do not cure or reverse its course. Newer therapies such as gene therapies, Rho-kinase inhibitors, and levodopa offer potential new targets for treatment.

## 1. Introduction

Age-related macular degeneration (AMD) is a degenerative disease of the retina and is the most prevalent retinal disease in the Western world, with the advanced form affecting 1–3% of its total population [1]. AMD accounts for 8.7% of all blindness worldwide [1] and is the most common cause of blindness in developed nations, with the highest prevalence among Europeans compared to African, Hispanic, and Asian populations. The estimated population of those suffering from AMD worldwide is 196 million in 2020, projected to increase to 288 million by 2040 due largely to increasing lifespan globally and Westernization of diet and lifestyle [1,2].

AMD is classified into two categories: a neovascular “wet” form and a non-neovascular “dry” form, or a mixture of both, all resulting in partial or complete loss of central vision [3]. While less common than dry AMD, wet or nvAMD accounts for almost 90% of blindness associated with AMD. Dry AMD currently has no treatment, and is characterized by thickening of Bruch’s membrane (BrM) due to lipid and protein accumulation, leading to the formation of sub-retinal pigment epithelium (RPE) deposits called drusen [4,5,6]. This deposition interferes with the fluid efflux from the RPE across Bruch’s membrane, chemically and mechanically separating the RPE from the choroid, further reducing perfusion to the RPE [7]. Additional strain is caused by oxidative stress, normal aging, inflammation, and lipofuscin/drusen accumulation, further reducing the flow of nutrients across the Bruch’s membrane [8]. RPE cell death then leads to the loss of photoreceptors, which ultimately leads to loss of vision. In this review, we will discuss etiology and pathophysiology common to AMD as a whole and then focus on those distinct to neovascular AMD plus highlight advances in research and varying treatment modalities for nvAMD to preserve ocular health.

## 2. Etiology of Neovascular AMD (nvAMD)

nvAMD is a multifactorial disease with numerous risk factors that are thought to play a role in its development. Associated non-modifiable risk factors include increased age, female sex, genetic factors, Caucasian race, and light iris color (Table 1). Modifiable risk factors include smoking, increased BMI, alcohol intake, and dietary habits [9,10,11,12].

Increasing age is the strongest demographic and leading risk factor associated with AMD, associated with increasing acellular deposits between the RPE and Bruch’s membrane [14,15]. Patients > 75 years of age had higher 15-year incidences of large drusen, RPE abnormalities, nvAMD, and geographic atrophy (GA) compared to people aged 43 to 54 years, with a 15-year cumulative incidence of late AMD in patients > 75 years at 8% [13]. A pooled meta-analysis of AMD in Europeans (25 studies, *n* = 57,173) shows a linear increase in the predicted prevalence of AMD, especially in patients > 60 years of age [13]. Progressive failure of repair mechanisms over a lifetime may explain this increased prevalence. One such example is atherosclerosis, the abnormal accumulation of cholesterol plaque within the vascular intima, for which increasing age is the dominant risk factor [5,16]. Subsequently, atherosclerosis is thought to contribute to AMD by atherosclerotic plaques altering choroidal circulation and venous drainage, therefore increasing blood flow resistance and actively increasing lipid deposition into Bruch’s membrane [17]. In support of this hypothesis, the Rotterdam study reported that atherosclerotic plaques in the carotid bifurcation in subjects below 85 years of age were associated with 4.7 times increased odds of developing and being affected by AMD. In addition, plaques in the common carotid artery as well as lower extremity arterial disease (ankle-arm index less than 0.90 on one side) showed a 2.5× increased prevalence developing AMD [18]. Progression of atherosclerosis is associated with hypertension (which is both modifiable and a disease of senesce) [19], with studies reporting that neovascular AMD was more positively associated with a diastolic blood pressure of >95 mm Hg (Odds ratio (OR) = 4.4) [12]. 

In addition to age, several studies report that race may also play a significant role in AMD (Table 2), with variations that occur in ethnic and geographic populations. Numerous studies have reported that Caucasians are more likely to develop AMD than blacks, including the Baltimore Eye Study, which reported a four-fold higher risk in whites than in blacks [20,21,22,23]. While Hispanics have been reported to be less susceptible to developing AMD, the overall prevalence of late AMD is 0.5%, and early AMD is increased in the Hispanic population [24]. A meta-analysis of 129,664 individuals among 39 studies reports that the prevalence of early and any AMD in Europeans was 11.2% and 12.3%, respectively, while that in Asians were 6.8% and 7.4%, and 7.1% and 7.5% in Africans, respectively [1].

Race and associated facial and pigment characteristics may alter the physiological and anatomical configuration, altering the likelihood of certain individuals developing nvAMD. Patients with darker irides have decreased nvAMD incidence, possibly due to increased melanin that absorbs greater light preventing light-induced ROS production and subsequent metabolic demand of the RPE. Caucasians with blue/hazel irides have a greater incidence of extensive macular disease and a greater visual field impairment than dark-colored irides in patients with unilateral nvAMD [29,30]. The Beaver Dam Eye Study found no correlation among Caucasians compared by hair color, skin sun sensitivity, iris color, and late AMD development at the 10-year follow-up [31]. Pigmentary characteristics on nvAMD incidence is controversial and hypothetical at this moment. In this hypothesis, Asians, who have light-colored skin should have a predisposition to developing nvAMD, however their incidence of nvAMD is reduced compared to Caucasians. This suggests that genetics may also play a vital part in nvAMD progression, with several single-nucleotide polymorphisms seen in nvAMD. The reader is asked to refer to Maguire et al. and DeAngelis et al. for a detailed review on the various genetic loci seen in nvAMD [32,33].

While a wide range of studies reveals the impact of intrinsic and genetic influences on nvAMD development, numerous environmental risk factors have been reported (Table 3). Pooled data from the Rotterdam, Beaver Dam Eye, and Blue Mountains studies report a positive correlation with a 2–4-fold likelihood of developing any type of AMD in smokers when compared to non-smokers [5,34,35,36,37,38], suggesting a pathological insult by increased choroidal vascular resistance [39] or oxidative damage to Bruch’s membrane [40,41]. To determine or measure the insult, a combined analysis of 14,752 participants from the Beaver Dam Eye Study, Rotterdam Study, and Blue Mountains Eye Study discovered that the development of nvAMD is greater (odds ratio 4.55) in smokers compared to non-smokers [42], with another study showing the time of onset of nvAMD occurs 5.5 years earlier in smokers than in never smokers [43]. It is hypothesized that smoke-induced damage in nvAMD is mediated through direct oxidation, depletion of anti-oxidants, activation of the immune system, as well as atherosclerotic vascular changes [44]. In mouse models, nicotine has been reported to induce pathologic angiogenesis, promoting choroidal neovascularization (CNV) [45]. This may be in part due to hypoxia that occurs after smoking, which stimulates the production of vascular endothelial growth factor VEGF, leading to neovascularization. Atherosclerosis or vasoconstriction secondary to smoking may further promote this cycle [41,46,47]. Studies exploring the effects of alcohol consumption and AMD development have shown that alcohol increases oxidative damage to the retina [48]. Numerous studies report an increased risk of nvAMD development due to alcohol consumption [49,50,51]. According to the 10-year follow up of the Beaver Dam Eye Study, a history of heavy drinking (four or more servings/day, one drink = 12-ounce beer, 4-ounce glass of wine, or 1.5 ounces of liquor) increased the likelihood of developing nvAMD, with a relative risk of 6.51 (adjusted for age and gender), than in non-heavy drinkers (0 servings/day) [37]. This suggests a dose-dependent relationship to the development of nvAMD based on the quantity or pattern of alcohol consumption. In contrast, the same researchers found on 15-year follow-up that there was no statistically significant association between heavy drinking and nvAMD development, but reported an increased risk of developing GA in heavy drinkers [52,53]. Like smoking, increased light exposure, and heavy drinking, body mass index also increases the body’s oxidative stress and is associated with early and late (nvAMD or GA) AMD [54]. For individuals in the obese group (≥30 kg/m^2^), studies report a 32% increase in the risk of developing late AMD. Additionally, the overweight group individuals (25–29.9 kg/m^2^) remained insignificant for its association with early and late AMD [55]. The Beaver Dam Eye Study reported increased BMI was associated with early AMD in female non-smokers (hazard ratio (HR) 1.10), and risk of late AMD among female non-smokers with increased with BMI (HR per 2.5 kg/m^2^ 1.31), waist to hip ratio (HR per 0.1 cm/cm 1.95), waist circumference (HR per 5 cm 1.21), and waist to height ratio (HR per 0.1 cm/cm 1.74) [56].

Together with BMI and other lifestyle choices, diet and nutrition are also thought to be modifiable risk factors (Table 3 and Table 4). The Mediterranean/Oriental diet lowers risks of nvAMD, while the Western diet increases the risk of nvAMD [2]. The Mediterranean and Oriental diets are characterized by high consumption of fruits, vegetables, legumes, whole grains, and seafood. These two regional diets have increased intake of micronutrients and lean meats while having reduced processed food in their diet; the Western diet pattern has a higher intake of red meat, processed meat, high-fat dairy products, fried potatoes, refined grains, and eggs. High adherence to the Mediterranean/Oriental diet was associated with a lower risk of advanced and nvAMD, while a high adherence to the Western diet showed an increased incidence for advanced AMD [2,65,66]. Higher intake of zinc, vitamin D, α-tocopherol, vitamin C, omega-3 fatty acids, and β-carotene was associated with a reduction of nvAMD by 60–90% with the Mediterranean and Oriental diets containing more nutritionally significant micronutrients than Western diets [2,67,68,69]. Dietary habits may also have a confounding impact on other factors such as body mass index and ethnicity, potentially leading to treatment modalities that target lifestyle modifications [70,71].

## 3. Pathophysiology

nvAMD’s pathophysiology is not fully understood; however, many studies have shown a multifactorial cause among genetic factors, metabolic factors, and environmental factors (Figure 1) [75,76]. Dry AMD is the precursor to wet AMD, with retinal dysfunction causing atrophic changes and subsequently leading to wet AMD [77]. The human retina is composed of roughly 91 million dim, light-sensitive rods, while the number of cones is 4.5 million, all with extremely high metabolic demand. The fovea is populated by a dense array of cones (200× the density in the fovea compared to the rest of the retina) [78]. Photoreceptor cells shed ~10% of their volume, and the outer photoreceptor segments (OS) basally regenerate roughly the same volume of cellular material each day, which are phagocytosed by the RPE and metabolically processed into waste products that are then ideally secreted through Bruch’s membrane into the choroidal circulation. Certain byproducts are more taxing to break down, such as lipofuscin [79]. Lipofuscin is a fluorescent material with significant phototoxic potential that is recognized as one of the hallmarks of aging [80] with three defining characteristics: they consist of intracellular secondary lysosomes, they emit yellow fluorescence when excited by near-ultraviolet or blue light, and they accumulate during normal senescence [81].

Lipofuscin in the retina is formed from the condensation of two molecules of all-trans-retinal and one molecule of phoshatidylethanolamine [82] and is believed to accumulate because of incomplete digestion of the shed OS in the RPE [83]. Once lipofuscin is in the RPE, it is converted to N-Retinylidene-N-Retinylethanolamine (A2E), a pyridinium bis-retinoid, which is toxic to RPE cells. RPE cells cannot dilute A2E by simple cellular division, so there is a progressive buildup of A2E within the cell, which inhibits phagolysosomal degradation of the OS by the RPE [84]. As the hallmark of AMD, drusen are amorphous, extra-cellular debris that accumulate with age in the area between RPE and the inner collagenous zone of Bruch’s membrane. They are categorized into two categories, “hard” and “soft,” depending on their size and shape. Few small hard drusen less than 63 µm can be found in at least 95% of the aged populations, but the presence of larger (≥125 µm) hard drusen, and more importantly, soft drusen (≥125–250 µm) in the macula is considered a major risk factor for developing advanced forms of AMD, including nvAMD [85].

Drusogenesis is a complex and multifactorial process that takes place over many years. The negative effect of drusen involves drusen formation between RPE and Bruch’s membrane, further decreasing perfusion from the choroid displacing the RPE and photoreceptor layer, effectively distorting vision. Drusen not only acts as a mechanical barrier but also activates the immune system, inducing local inflammation. Drusen is primarily composed of esterified cholesterol (EC), phosphatidylcholine (PC), and protein. Major proteins include vitronectin, complement component 9, apolipoprotein E, clusterin, ATP synthase subunit β, scavenger receptor B2, and retinol dehydrogenase 5 [86]. The immune-associated components include dendritic cell processes [87], major histocompatibility complex (MHC) class II antigens [88], immunoglobulins [89], and components of the complement cascade [90]. Recent studies support the hypothesis that deposits between Bruch’s membrane and the RPE may act as a stimulus for the complement system’s local activation. Due to the strong chemotactic activity of complement activation products, there is an influx of inflammatory cells, which may cause further growth of the deposits and subsequent RPE oxidative cell death. The extra-cellular deposits also contribute to local ischemia of RPE cells, which release angiogenic stimuli leading to choroidal neovascularization [91].

Neovascular AMD, wherein abnormal vessels break through Bruch’s membrane (Figure 2) is one of the leading causes of blindness in the West. It is mediated through several mechanisms. Oxidized low-density lipoproteins, which can exacerbate the deposits in the RPE, inhibit the breakdown of OS in the RPE by delaying the maturation of RPE phagosomes. This leads to the inefficient degradation of the OS and leads to a build-up of undigested lipids and proteins within the RPE, resulting in further production of compounds like A2E [92]. A2E-treated ARPE19 cells increase the expression of pro-angiogenic factors and decreases the expression of anti-angiogenic factors both in vitro and in vivo, with CNV activity greater in A2E-treated eyes [93]. The accumulation of photo-oxidized products in the RPE is believed to be an underlying cause of age-related macular degeneration [79].

With the retina being one of the highest oxygen-consuming tissues in the body, it is particularly susceptible to oxidative stress because of its high consumption of oxygen, a high proportion of polyunsaturated fatty acids, and its exposure to visible light [94]. The macular pigment acts as a natural barrier protecting the central retina against oxidative damage. The macular pigment (MP) contains three main dietary carotenoids; lutein, zeaxanthin, and meso-zeaxanthin, which forms a natural barrier protecting the retina against oxidative damage, as well as protecting the retina via acting as an optical filter that absorbs short-wave visible light [95,96,97]. Reduced intake of carotenoids and smoking is shown to reduce these macular pigments; however, the association with MP reduction and smoking is not yet clear [98,99]. Still, smoking is a key risk for nvAMD development.

While the MP protects the retina from light oxidation, the shed photoreceptor outer segment membranes are rich in PUFA and can be readily oxidized to induce a cytotoxic chain reaction. The OS segments are degraded in the RPE phagolysosomes, which are not completely digested and aggregate as lipofuscin, deposited in RPE cells’ lysosomes [97]. A2E-loaded RPE cells generated high levels of both hydrogen peroxide (H_2_O_2_) and superoxide anion (O_2_•−) when exposed to blue-violet light [100], mice deficient in superoxide dismutase develop features of AMD including CNV [101]. Factors like cigarette smoke increase oxidative stress by generating ROS and reducing anti-oxidant capacity. Cigarette smoke has been shown in a mouse model to form sub-RPE deposits, thickening of Bruch’s membrane, and accumulation of deposits within the Bruch’s membrane. Normally, oxidative damage is minimized by the presence of anti-oxidant damage and repair systems [102], but anti-oxidant systems are affected with smoking, with dietary anti-oxidants such as carotenoids, retinoids, and tocopherols reduced in smokers (after adjusting for dietary intake), but the mechanism behind this reduction is not yet clear [34,35,36,37,38,103,104,105]. Aging also reduces restorative/ROS coping systems, with increased oxidative damage, including mitochondrial DNA that is more susceptible and essential to the high metabolic demand of RPE [106]. Aging also increases the deposition of beta-amyloid proteins in the eyes. Amyloid proteins are a known constituent of drusen, which can further cause progression to nvAMD [107,108].

Another type of cell damage or death is induced by caspases, which are cysteine proteases that initiate cellular programs leading to inflammation or cell death [109]. They can either be pro-inflammatory or proapoptotic, depending on the domain architecture and the described functions [110]. Caspase-1 is produced as an inactive zymogen that is recruited by multi-protein complexes called inflammasomes [111], which act as molecular platforms activated by cellular stress. They function in the innate immune response for activation of caspase-1, and subsequent secretion of active interleukins (IL-18 and IL- 1β), which mediate the inflammatory response [112,113,114]. The NLRP3 inflammasome is a critical component of the innate immune system that is linked with AMD (Figure 3) [115].

IL-18 and IL-1β are synthesized as inactive precursors that require cleavage by caspase-1 to produce biologically active cytokines [117]. IL-1β is considered a classic activator and mediator of inflammation [118] and functions as a potent pro-angiogenic factor by stimulating VEGF production [119]. IL-18 can stimulate both Th1 and Th2 responses depending on the local microenvironment [120,121] and has been implicated in several inflammatory conditions. IL-18 function has been theorized to have a protective effect in other disease models [122]. IL-18 can play either a protective and/or a pro-inflammatory role, depending on the cell’s immunological status or the type and phase of the inflammatory process [123].

With drusen and lipofuscin being essential components of nvAMD development, it has been shown that drusen components, as well as A2E, could activate NLRP3 leading to the activation of the caspase-1 cascade [124,125]. It has been reported that NLRP3 activation can also be caused by oxidative stress and VEGF-A expression [126,127,128]. In addition, studies report that a decrease in extra-cellular osmolarity induced a K(+)-dependent conformational change of the preassembled NLRP3-inactive inflammasome during cell swelling, followed by activation of NLRP3 [129], leading to the cascade of caspase activation and resulting inflammatory response to the retina. Chronic inflammation also leads to P2 × 7R activation by ATP released from damaged retinal cells. P2X7R also directly activates the NLRP3 inflammasome causing further inflammation [130].

NLRP3 was also activated in GA in response to repetitive Alu RNA accumulated in RPE due to a loss of DICER1. Alu RNA was also required for activation of caspase-1 in the inflammasome complex [131,132]. Alu RNAs are short (approximately 300 base pairs) interspersed nuclear elements transcribed from Alu elements in retrotransposons and are the most abundant repetitive elements in the human genome [133]. DICER1 is an RNase that processes double-stranded and self-complementary RNAs, including a majority of premature micro-RNAs (miRNAs) into their bioactive forms [134,135]. DICER1 also metabolizes Alu RNAs in humans with DICER1 deficiency implicated in RPE cell death in nvAMD due to accumulation of unprocessed Alu RNAs, resulting in activation of the NLRP3 inflammasome cascade [132,136]. Wright et al. reported that genetic suppression of DICER1 in three independent mouse models manifested as focal RPE atrophy and aberrant CRNV (choroidal and retinal neovascularization) and that DICER1 expression is reduced in a mouse model of spontaneous CNV. DICER1 as a means of treatment has also been reported, with reports that Aden-associated virus (AAV)-enforced expression of a DICER1 construct successfully escaped miRNA negative feedback, reducing spontaneous CNV in mice [137]. The RPE is crucial to nvAMD development, and a pro-inflammatory retinal environment is promoted by RPE response to various stresses modulating CNV development and progression. The complement system here plays an important role in creating a pro-inflammatory retinal milieu with complement factors C3a and C5a inducing RPE secretion of VEGF and being potent chemotactic agents by recruiting leukocytes to the choroid [138].

Oxidative stress to the RPE by photo by-products such as lipofuscin activates complement, and the induced autoimmune reaction causes complement deposition in the retina [138,139].

There are multiple pathways the RPE can regulate the retinal immune-landscape and, therefore, neovascularization in AMD [140]. In CNV, the macrophage is attracted to the retina with large numbers of retinal macrophages being a hallmark of CNV. However, it could be debated that the increase of macrophages may represent an exacerbation of disease vs. a compensatory vascular-dampening response [140]. In support of macrophages being pro-angiogenic, inhibition of monocyte migration to the retina reduced CNV in a laser-induced mouse model of the disease [141,142]. In contrast, in a non-injury mouse model of AMD, mice that possess defects in macrophage mobilization develop choroidal neovascularization, suggesting that macrophages somehow also protect against CNV [143]. Given the available evidence, macrophages’ most likely role in CNV is determined by local macrophage-polarizing factors. Undifferentiated M0 macrophages can differentiate into the classical (M1) or alternate (M2) pathway. M1 pathway is pro-inflammatory, and the M2 pathway is anti-inflammatory with stimulation of the M2 macrophages, showing increased angiogenic potential compared to M1 macrophages [144,145,146,147]. The uncertainty of macrophages’ role in CNV is also unclear following anti-vascular treatment, as bevacizumab increases the number of retinal macrophages within human neovascular membranes [148].Walshe et al. showed that blockage of VEGF-A increased leucocyte-endothelial adhesion (LEA), which could increase retinal macrophage following anti-VEGF treatments [149], as well as compensatory elevation in VEGF-A levels following anti-VEGF therapy [150]. This warrants further study on the impact and role of retinal macrophages in angiogenesis in AMD.

Another theory of neovascularization is angiopoietin vascular growth factors that activate pathways in the retinal vasculature. Tyrosine kinase with immunoglobulin-like and epidermal growth factor-like domains 2 (Tie2) is a transmembrane receptor that serves as a binding site for hormones angiopoietin 1 (Ang1) and angiopoietin 2 (Ang2) ligands [151]. Ang1 is a full endogenous agonist of Tie2 produced in pericytes surrounding the retinal vasculature. Once bound to Tie2, it phosphorylates the receptor, activating downstream pathways that suppress vascular permeability and maintain vascular stability [152]. Conversely, Ang2 is an endogenous weak, partial agonist of Tie2 that competes with Ang1 to suppress phosphorylation and leads to the activation of Tie2. Overexpression of Ang2 and Tie2 antagonism in animal models results in incorrect vessel organization and failure of vessels to mature [153]. Ang1 co-expressed with VEGF-A prevented retinal detachment and blocked neovascularization [154], while Ang2 is believed to play a role in angiogenesis, vessel destabilization, and inflammation [155,156]. Elevated Ang2 was also found in the aqueous humor of patients with nvAMD and in the vitreous of diabetic patients undergoing a vitrectomy, further adding proof of its pro-inflammatory and pro-angiogenic nature [157,158]. Farcizumab is a new anti-VEGF injection that blocks VEGF as well as Ang2, which is currently being tested [159].

Vascular endothelial growth factor is a potent proangiogenic factor first thought to be an endothelial cell specific mitogen [160], with physiological functions including bone formation [161], hematopoiesis [162], wound healing [163], and development of the vascular and lymphatic system [164,165]. VEGF is upregulated in many tumors and is one of the contributing factors of tumor angiogenesis [166]. VEGF-A, VEGF-B, VEGF-C, VEGF-D, and VEGF-E and placental growth factor (PGF) are members of the VEGF family [167,168], with VEGF-A strongly inducing vascular proliferation and migration of endothelial cells essential for both physiological and pathological angiogenesis [169]. The production of VEGF can be induced in hypoxic cells; when cells are in a low-oxygen environment, they produce the transcription of the hypoxia-inducible factor 1 alpha (HIF-1 alpha) inducing the release of VEGF-A [170]. VEGF-A acts via endothelial-specific receptor tyrosine kinases (VEGFR1, VEGFR2) located in endothelial cells, with VEGFR-2 being the most important for angiogenesis [171,172]. Oxidative stress triggers many other pathways, with the autocrine signaling and upregulation of VEGF being no exception. Oxidative stress and ROS synergize with other pathways (NLRP3, complement cascade, Ang 2, hypoxia), further promoting angiogenesis and progression of nvAMD [173,174]. Once VEGF-A binds to the receptor, several signaling pathways are activated: (1) the Mitogen-activated protein kinase- (MAPK-) p38 signaling pathway, where the effector heat shock protein (HSP27) reorganizes actin [175], (2) the phosphatidylinositol 3-kinase- (PI3K-) AKT protein kinase B pathway, promoting the formation of nitric oxide [176,177], and (3) the phospholipase C gamma (PLC*γ*) releasing intracellular calcium, promoting prostaglandin production, increasing vascular permeability [178,179]. Rearrangement of the cytoskeleton is crucial for various steps of angiogenesis, with actin being a key cytoskeletal element integral for cell division and motility; actin rearrangement leads to formation of protrusive structures, generating intracellular forces required for cell migration [180,181,182]. Nitric oxide is a known vasodilator that can lead to upregulation of VEGF by enhancing HIF-1 alpha activity under normoxic conditions [183,184].

Under healthy conditions, the production of VEGF upregulates the production of PEDF [185] (pigment epithelium-derived factor) via the RPE [186]. PEDF is a potent endogenous down-regulator of angiogenesis [187], via intracellular translocation of the transmembrane domain of VEGFR-1 [188]. With RPE dysregulation being a key feature of AMD [189], the protective role of PEDF is reduced, with studies reporting a decrease in PEDF in the choroid of AMD eyes, suggesting that decreased PEDF in the choroid-BrM-RPE [190] creates a non-restrictive environment for the progression to nvAMD [191].

In nvAMD, anti-VEGF therapy is currently the only treatment available, which only slows down the progression of the disease, however more knowledge about the downstream effects of factors such as VEGF may offer another treatment modality.

## 4. Treatment

The treatment modalities for nvAMD have greatly evolved over time. The earliest form of nvAMD treatment with laser photocoagulation showed reduced loss of visual acuity (VA) with severe VA loss in 60% of untreated eyes, compared to 25% of eyes treated with argon laser photocoagulation [192]. Krypton red laser photocoagulation for CNV slightly reduced VA loss of six or more lines at three years in 49% of treated eyes vs. 58% of untreated eyes [193]. Photodynamic therapy (PDT) with verteporfin was a predominant treatment for sub-retinal neovascular membrane (SNRVM). When irradiated with a 689 nm laser light, verteporfin releases radical oxygen species that damage the choroidal neovascular endothelium [194,195], circumventing the adjacent’s damage retina and resulting in short term cessation of fluorescein leakage from choroidal neovascularization [196,197]. Through extensive clinical trials (Table 5), anti-VEGF compounds have drastically changed treatment for nvAMD. The first anti-VEGF agent approved for use in nvAMD was Pegaptanib (Macugen) [198]. Since then, newer anti-VEGF therapies have emerged and widely replaced the use of Pegaptanib, such as brolucizumab (Beovu), ranibizumab (Lucentis), aflibercept (Eylea), and the off-label use of bevacizumab (Avastin) [199]. Anti-VEGF therapy is the mainstay of wet AMD treatment as it affects multiple key pathogenic pathways (Figure 4). Apart from nvAMD, anti-VEGF therapy has been used for other ocular (diabetic macular edema [200], vascular occlusions [201], vitreous hemorrhage [202], severe diabetic retinopathy [203], neovascular glaucoma [204], Eales disease [205], Coats disease [206], iris neovascularization [207], post-operative trabeculectomy [208], pterygium [209], retinopathy of prematurity [210], post-operative cystoid macular edema [211]) and oncological diseases (ovarian cancer [212], gastric cancer [213], prostate cancer [214], GIST(Gastrointestinal stromal tumor) [215], pancreatic cancer [216], non-small cell lung carcinoma [217]).

Ranibizumab (Lucentis) is a 48-kD Fab fragment of the A4.6.1 antibody. The A4.6.1 antibody is one of the four antibodies of the IgG1 isotope that most effectively binds and neutralizes different isomers of VEGF-A [233,234]. Bevacizumab (Avastin) is a full-length monoclonal antibody (149 kDa) that binds to all isoforms of VEGF-A [235]. It was approved by the FDA in February 2004 for the treatment of metastatic colorectal cancer, and is currently the cheapest of the anti-VEGF drugs [236]. Aflibercept is a 115 kDa fully human, a recombinant fusion protein composed of immunoglobulin binding domain of VEGF receptor 1 and 2, fused to the Fc region of human IgG that binds to VEGF-A, VEGF-B, and placental growth factor (PlGF). In addition to its use as a nonirritating intravitreal injection, it is also used for the treatment of cancer. Aflibercept uniquely binds to PlGF, which is present in human CNV membranes and contributes to the development of experimental CNV [237,238]. The loss of PIGF signaling seen in aflibercept is also useful in treatment of diabetic retinopathy, as aflibercept leads to blockage of the ERK pathway and subsequent TNF-alpha suppression in high glucose states [239]. Brolucizumab (Beovu) is a humanized single-chain variable fragment (scFv) that is a potent inhibitor of VEGF-A. The scFv comprises the monoclonal antibody’s variable light and heavy chain domains tethered by a flexible linker, resulting in a small fragment weighing approximately 26 kDa, which increases bioavailability, reduces immunogenicity, and allows better ocular tissue penetration [229,240,241]. Brolucizumab is the newest anti-VEGF drug on the market, and aims to reduce the number of annual anti-VEGF injections. The HAWK and HARRIER trials compared 3 mg and 6 mg of brolucizumab with 2 mg of aflibercept spaced at 12 weeks (brolucizumab) and 8 weeks (aflibercept), with a reduction to q 8 weeks (brolucizumab) if necessary. In brolucizumab-treated eyes on the q12w regimen at the week 44 in HAWK, there was an 80.5% (3 mg) and 81.5% (6 mg) probability of remaining on an q12w interval until week 92, with a 75.4% for the brolucizumab 6 mg group in HARRIER. The probability of maintaining the q 12-week regimen from loading to week 92 was 39.7% and 45.4% for the brolucizumab 3 mg and 6 mg treatment groups in HAWK, and 38.6% for the 6 mg group in HARRIER [229]. Pluykhova et al. compared the safety profile of long-term use of ranibizumab, bevacizumab, and aflibercepts in the RCT (randomized clinical trial) meta-analysis, with the study hypothesizing the potential benefits of ranibizumab vs. bevacizumab in terms of a systemic safety profile. However, more direct RCT on ocular safety needs to be conducted to validate these findings further [242].

### Future Advancements

Conbercept (143 kDa) is an anti-VEGF agent that consists of the extra-cellular domain 2 of VEGFR1 and extra-cellular domains 3 and 4 of VEGFR2 fused to the Fc portion of human IgG1. Similar to aflibercept, conbercept targets VEGF-A, VEGF-B, and PIGF [243,244]. Farcizumab is the first bispecific antibody for both VEGF-A and angiopoietin 2 (Ang-2) specifically designed for intraocular use assembled using Roche’s CrossMAb technology, which binds to both VEGF-A and Ang-2 with high affinity and specificity. It has been engineered to abolish certain binding interactions to Fc ϒR and Fc Rn, which increased systemic but not the ocular clearance [245]. Patients treated with farcizumab had a greater mean BCVA increase that was maintained at week 52, with corresponding anatomic improvements compared to ranibizumab [232].

Ripasudil is a Rho-associated kinase inhibitor developed originally for the treatment of glaucoma and ocular hypertension [246]. Rho-associated protein kinase (ROCK 1 and ROCK 2) determine the macrophage polarization into the M1 and M2 subtypes. Aging increases ROCK2 signaling, resulting in over-expression of the pro-angiogenic macular degeneration associated macrophages, which increases IL-4 leading to angiogenesis [145,146]. A recent study showed that ripasudil suppressed expression levels of ROCK1, ROCK2, and miR-136-5p, resulting in inhibition of NLRP3, ASC, caspase1, IL-1β, and IL-18 [247].

Gene therapies targeting nvAMD have been performed on animal models, and small-scale clinical trials have been performed on human subjects. Soluble fms-like tyrosine kinase-1(sFlt-1) is an anti-angiogenic receptor for VEGF that binds and causes VEGF sequestration [248]. Animal models of rAAV.sFLT-1 injected mice caused upregulation of sFlt-1, reduced retinal neovascularization [249]. In human studies, small scale r.AAV.sFLT-1 have been conducted with inconclusive results [250,251]. There are many other vectors/targets being used in gene therapy (Table 6), but gene therapy is currently in its infancy [252,253,254,255,256]. With the immense potential of gene therapy to treat multiple diseases, more research into the vectors and their targets is needed to further validate possible therapeutic use in nvAMD.

Levodopa’s effects on nvAMD have been studied on the basis that the RPE has a G protein-coupled receptor (GPR143) activated by levodopa. As part of normal aging, melanin, as well as pigment epithelium-derived factor (PEDF), are reduced, and GPR413 activation via signaling reduces VEGF secretion and simultaneously increases PEDF. Studies show that patients using levodopa have a lower risk and a greater age of onset of nvAMD, with Figueroa et al., reporting that levodopa delayed anti-VEGF therapy while improving visual outcomes. In the first month, retinal fluid decreased by 29% without anti-VEGF treatment, with the retinal fluid sustaining for six months. Mean visual acuity improved by 4.7 and 4.8 letters in cohort-1 (levodopa, newly diagnosed, naïve anti-VEGF) and cohort-2 (levodopa, previous anti-VEGF) with a 52% reduction in the need for anti-VEGF injections in Cohort-2 [261].

## 5. Conclusions

AMD is one of the most common retinal disorders that affect millions, predominantly in the Western world. It is a disease of the developed world with worldwide implications. nvAMD is one subtype that causes 90% of blindness associated with AMD. With the abundant research being conducted, our understanding of nvAMD and the pathologic mechanisms has progressed along with a wide array of treatment modalities, with newer agents being investigated. Anti-VEGF therapy remains the mainstream treatment for slowing the disease’s progression, but is expensive, requires frequent treatments and ultimately does not treat the disease entirely. Due to the success of anti-VEGF therapy, newer anti-VEGF drugs are being research to further increase the treatment efficacy and to reduce cost by prolonging the treatment interval. Due to its multifactorial nature, patients must be educated about modifiable risk factors and the lifestyle changes that can prevent or delay the development of the disease in susceptible individuals. Treatment modalities are evolving that target inflammatory processes, signaling proteins, gene therapies, and newer drugs are currently under trial to treat/delay the disease or prolong the treatment interval.

## Figures and Tables

**Figure 1 ijms-22-01170-f001:**
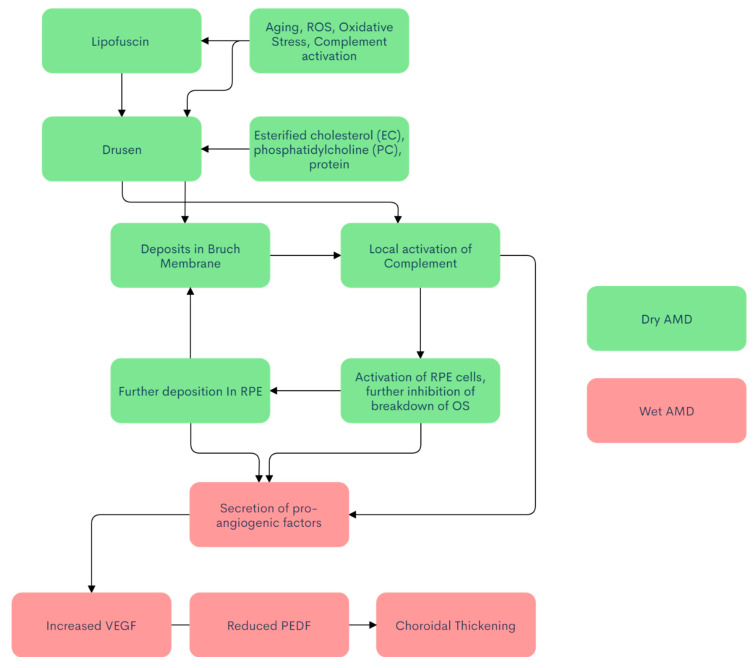
Flowchart of AMD pathogenesis.

**Figure 2 ijms-22-01170-f002:**
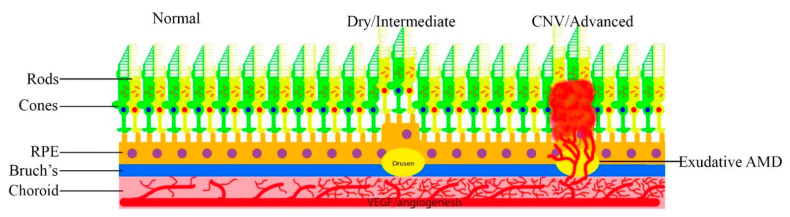
Progression of AMD.

**Figure 3 ijms-22-01170-f003:**
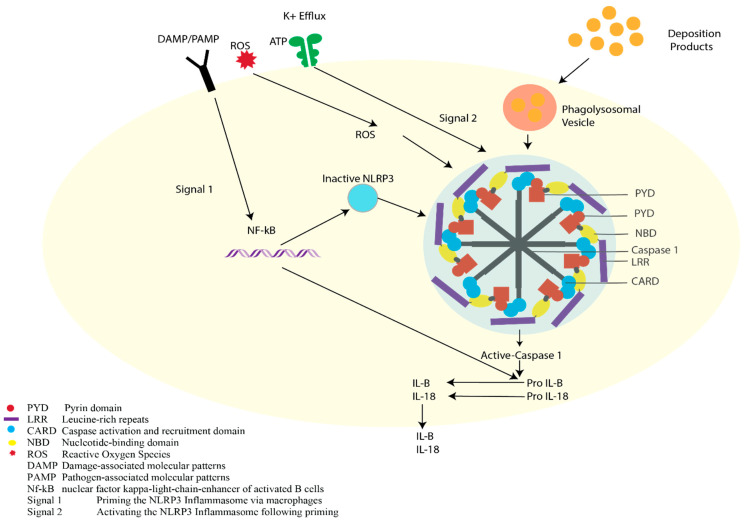
NLRP3 inflammasome [116].

**Figure 4 ijms-22-01170-f004:**
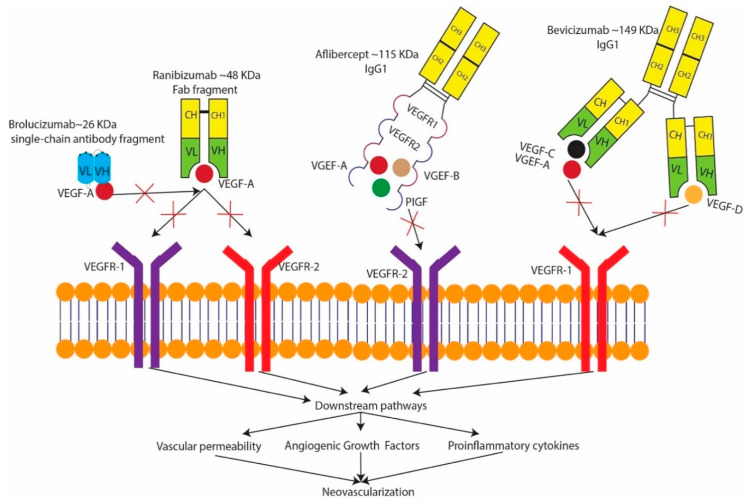
Mechanism of anti-VEGF drugs.

**Table 1 ijms-22-01170-t001:** Estimated prevalence according to age and neovascular age-related macular degeneration (nvAMD) in populations of European ancestry.

Age (Years)	Predicted Prevalence % (nvAMD)	Range
50	0.04	0.02–0.07
55	0.08	0.05–0.13
60	0.17	0.11–0.25
65	0.34	0.23–0.49
70	0.7	0.48–0.97
75	1.4	0.99–1.90
80	2.79	1.99–3.79
85	5.48	3.91–7.45
90	10.49	7.45–14.37

Prevalence estimates are based on using International Classification or Wisconsin Age-Related Maculopathy Grading system together with fundus photography/imaging. Data obtained from Rudnicka et al [13].

**Table 2 ijms-22-01170-t002:** nvAMD prevalence between different races.

Study	Participants (*n*=)	Population	nvAMD Prevalence
Beaver Dam Eye Study [25]	4771	White	1.2% Total
National Health and Nutrition Examination Survey III [26]	7888	White	0.30%
		Black	0.10%
		Mexican-Americans	0.10%
The Baltimore eye Study [20]	4361	White	0.60%
		Black	0.10%
The Los Angles Latino Eye Study [27]	5875	Mexican-Americans	0.29%
Proyecto [24]	2776	Mexican-Americans	0.14%
The Salisbury Eye Evaluation Project [21]	2008	White	1.70%
		Black	1.00%
Hisayama Study [28]	1486	Japanese	0.67%

**Table 3 ijms-22-01170-t003:** Modifiable risk factors for AMD.

Factor	Characteristics	Increased Risk	Suggested Method of Action
Smoking	Current smokers	4.55 fold nvAMD * [57]	Direct oxidation of RPE [4]
			Depletion of antioxidants [58]
	Former smokers	1.54 fold nvAMD * [57]	Activation of Complement [59]
		* compared to having never smoked	Atherosclerotic vascular changes [60]
Alcohol	Heavy drinker (daily alcoholic beverages >4)	RR 6.51 nvAMD * [37]	Oxidative stress [61]
		* compared to subjects with 0 g Alcohol/day, adjusted for age and gender	
BMI	Obesity (≥30 kg/m^2^)	32% increase incidence of late AMD [55]	Oxidative stress [62]
Diet	Mediterranean diet	−26% lower risk of progression to advanced (nvAMD or GA) AMD [2]	Certain foods are retino-protective or rich in anti-oxidants [63,64]
	Western diet	+56% increase in early AMD	
		+270% increase advanced AMD [2]	

**Table 4 ijms-22-01170-t004:** Nutritional and dietary factors in AMD.

Study Authors	Diet Rich vs. Poor in Nutrient	Dose Required for Reduction	Odds Ratio	Reduction in AMD
Aoki et al., (2016) [72]	Zinc	≥10.2 (mg/d)	0.10 ^1^	nvAMD = 60–90% [2]
	Vitamin D	≥27.5 (μgram/d)	0.40 ^1^	
	α-tocopherol	≥10.6 (mg/d)	0.20 ^1^	
	Vitamin C	≥184.9 (mg/d)	0.40 ^1^	
	Omega−3 fatty acids	≥3.9 (g/d)	0.20 ^1^	
	β-carotene	≥6.04 (mg/d)	0.20 ^1^	
Hogg et al., (2017) [73]	MDS * score	Quartile 4 (high)	0.53 ^2^ 0.61 ^3^	nvAMD = 47% [2] nvAMD = 39% [2]
SanGiovanni et al., (2007) [74]	Total fish intake	>2 servings/week		nvAMD = 39% [2]
Chiu et al., (2014) [66]	Oriental vs. Western diet	Diet quintile		
	Oriental pattern score	Quintile 5 (high)	0.38 ^4^	Advanced AMD = 62% [2]
	Western pattern score	Quintile 5 (high)	3.7 ^5^	Advanced AMD = 270% increase [2]

* MDS = Mediterranean diet score; ^1^ ORs compare the highest (Q5) intake of micronutrients to lowest (Q1); ^2^ ORs compare the highest (≥6) to lowest adherence (≤4) of the MDS; ^3^ ORs compare >2 medium servings/week with controls for levels (<1 medium serving/month) of fin fish and shellfish intake; ^4^ ORs compare the highest to lowest quintile of the Oriental pattern score; ^5^ ORs compare the highest to lowest quintile of the Western pattern score.

**Table 5 ijms-22-01170-t005:** Anti-vascular endothelial growth factor (Anti-VEGF) clinical trials.

Study Name	Drug	Control	Treatment	Follow up	Change in ETDRS	
					Treatment ^1^	Control ^2^
MARINA [218]	Ranibizumab	Sham injections q 4 weeks	Ranibizumab 0·5 mg q 4 weeks	1 year	+7.2	–10.4
ANCHOR [219]	Ranibizumab	Sham injection q 4 weeks plus PDT	Ranibizumab 0·5 mg q 4 weeks+ sham PDT	1 year	+11.3	–9.5
HARBOR [220]	Ranibizumab	Ranibizumab 0·5 q 4 weeks	Ranibizumab 0·5 mg q 4 weeks × 3 months then as required	2 years	+7.9	+9.1
TREX [221]	Ranibizumab	Ranibizumab 0·5 q 4 weeks	Ranibizumab 0·5 mg q 4 weeks × 3 months until disease inactive then extension per protocol	1 year	+10.5	+9.2
FOCUS [222]	Ranibizumab	Sham injections+PDT	Ranibizumab+PDT 0.5 mg q 4 weeks, PDT = day one + quarterly if needed	2 years	4.6	−7.8
BRAMD [223]	Bevacizumab	Ranibizumab 0·5 mg q 4 weeks	Bevacizumab 1·25 mg q 4 weeks	1 year	+5.1	+6.4
LUCAS [224]	Bevacizumab	Ranibizumab 0·5 mg q 4 weeks until nvAMD is inactive, then extension by 2 weeks (maximum of 12 weeks)	Bevacizumab 1·25 mg q 4 weeks until nvAMD is inactive, then extension by 2 weeks (maximum of 12 weeks)	1 year	+7.9	+8.2
CATT [225]	Bevacizumab	Ranibizumab 0·5 mg q 4 weeks	Bevacizumab 1·25 mg q 4 weeks	1 year	+7.8	+8.8
GEFAL [226]	Bevacizumab	Ranibizumab 0·5 mg q 4 weeks × 3 months, then as needed	Bevacizumab 0·5 mg q 4 weeks × 3 months, then as needed	1 year	+4.8	+2.9
IVAN [227]	Bevacizumab	Ranibizumab 0·5 mg q 4 weeks	Bevacizumab 1·25 mg q 4 weeks	2 yearss	+4.1	+4.9
VIEW 1 [228]	Aflibercept	Ranibizumab 0·5 mg q 4 weeks	Aflibercept 2 mg every 2 months	1 year	+7.9	+8.1
VIEW 2 [228]	Aflibercept	Ranibizumab 0·5 mg q 4 weeks	Aflibercept 2 mg every 2 months	1 year	+8.9	+9.4
HAWK [229]	Brolucizumab	Aflibercept 2 mg q 8 weeks	Brolucizumab 3 mg q 12 weeks Or Brolucizumab 6 mg q12 weeks	96 weeks	+5.6 +5.9	+5.3
HARRIER [229]	Brolucizumab	Aflibercept 2 mg q 8 weeks	Brolucizumab 6 mg q12 weeks	96 weeks	+6.1	+6.6
AURORA [230]	Conbercept	Conbercept 0·5 mg q 4 weeks × 3 months, then monthly or as required	Conbercept 2 mg q 4 weeks × 3 months, then monthly or as required	1 year	+15.4	+9.3
PHOENIX [231]	Conbercept	Sham injections q 4 weeks × 3 months then conbercept 0.5 mg q 4 weeks × 3 months	Conbercept 0.5 mg q 4 weeks × 3 months, then quarterly	1 year	+9.98	+8.81
STAIRWAY [232]	Farcizumab	0.5 mg ranibizumab q 4 weeks × 52 weeks	Farcizumab 6 mg q 16 weeks	52 weeks	+11.4	+9.6
			Or			
			Farcizumab 6 mg q 12 weeks		+10.1	

1 = Letters gained with treatment; 2 = Letters gained/lost from baseline control.

**Table 6 ijms-22-01170-t006:** Ongoing gene therapy clinical trials for nvAMD.

Trial Name	Targeted Gene	Vector	Sponsor
NCT01494805 [257]	sFLT01	AAV-2	Adverum Biotechnologies, Inc. (Redwoodcity, CA, USA)
NCT03585556 [258]	sCD59	AAV-2	Hemera Biosciences (Waltham, MA, USA)
NCT01024998 [254]	sFLT-1	AAV-2	Sanofi Genzyme (Cambridge, MA, USA)
NCT03748784 [259]	Aflibercept	AAV-2	Adverum Biotechnologies, Inc. (Redwoodcity, CA, USA)
NCT03066258 [253]	Anti-VEGF Fab	AAV-8	Regenxbio Inc.(Rockville, MD, USA)
NCT01678872 [256]	Endostatin/Angiostatin	EIAV	Oxford BioMedica (Oxford, UK)
NCT00109499 [255]	PEDF	AAV5	GenVec (Gaithersburg, MD, USA)
NCT03999801 [260]	Monoclonal antibody fragment	AAV-8	Regenxbio Inc. (Rockville, MD, USA)

## Data Availability

No new data were created or analyzed in this study. Data sharing is not applicable to this article.

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
