# Peer review of "Neovascular Macular Degeneration: A Review of Etiology, Risk Factors, and Recent Advances in Research and Therapy"

_ijms, 2021, doi:10.3390/ijms22031170_

Round 1

Reviewer 1 Report

In this review authors have described a timely and interesting topic focusing on the new therapeutic strategies that are suitable in contrasting the neovascular injury in retina degeneration. However, a low number of issues need to be addressed. Authors need to deepen some aspects and update bibliographic references. Moreover, they need to slightly fix few points in the text.

Comments:

  1. Page 9 lines 250-253: “In addition, studies report that a decrease in extra cellular osmolarity induced a K(+)-dependent conformational change of the preassembled NLRP3- inactive inflammasome during cell swelling, followed by activation of NLRP3[128], leading to the cascade of caspase activation and resulting inflammatory response to the retina ”. Please see and cite the following paper:

  • Platania CBM, et al. P2X7 receptor antagonism: Implications in diabetic retinopathy. Biochem Pharmacol. 2017; 138:130-139.

  1. Page 8 lines 225-227 Aging also reduces restorative/ROS coping systems, with increased oxidative damage, including mitochondrial DNA that is more susceptible and essential to the high metabolic demand of RPE”. Please see and cite the following paper:
  • Fisichella V,et al. TGF-β1 prevents rat retinal insult induced by amyloid-β (1-42) oligomers. Eur J Pharmacol. 2016; 787:72-7.

  1. Page 9 lines 229-231 They can either be pro- inflammatory or proapoptotic, depending on the domain architecture and the described functions “. Please see and cite the following paper:
  • Lazzara F, et al. Aflibercept regulates retinal inflammation elicited by high glucose via the PlGF/ERK pathway. Biochem Pharmacol. 2019; 168:341-351.

  1. Regarding the section on the “4. Treatment” a direct comparison of RBZ, BVZ and AFB         safety profiles in the RCT network meta-analytical setting have not revealed a consistent benefit of these three commonly used anti-vascular endothelial growth factor (anti-VEGF)    agents in AMD. Network model ranking highlighted potential benefits of RBZ in terms of a         systemic safety profile; however, this appears a hypothesis rather than a conclusion. The authors should discuss more in details these aspects.

            Please see and cite the following paper:

            Plyukhova AA, et al. Comparative Safety of Bevacizumab, Ranibizumab, and Aflibercept for            Treatment of Neovascular Age-Related Macular Degeneration (AMD): A Systematic Review            and Network Meta-Analysis of Direct Comparative Studies. J Clin Med. 2020 May           18;9(5):1522.

  1. Also in the paragraph on Anti_VEGF also briefly describe all their actual indications. It is usefull for readers to have a more complete overview on the topic.
  1. Conclusions: Conclusions needs to be underlined more incisively.

Reviewer 2 Report

The authors present an updated review about the pathogenesis, current and future treatment options for neovascular AMD (as proposed in the title). Nevertheless already multiple reviews have been published about this subject, the review has a high impact due to the mass of ongoing research, new knowledge and drugs. Additionally, in spite of ongoing research, dry AMD still remains untreatable and progression of wet AMD can only be slowed down and treated symptomatically by expensive and logistically challenging intravitreal injections of anti-VEGF molecules.

However, the manuscript does not clearly differentiate between the two forms of AMD and misses some key information why a careful revision is recommended.

Major comments

  1. AMD vs. dry AMD vs. nvAMD

In the whole manuscript the forms and the terms “AMD”, “dry AMD” and “nvAMD” are not clearly differentiated and often even mixed (e.g. in the text in lines 45-51 risk factors of AMD are described and the authors refer to table 1; however, in table 1 only nvAMD is mentioned). Moreover, the chapters introduction, etiology and pathophysiology are focused on dry AMD and nvAMD is not or only mentioned briefly, though the title promises a review especially about nvAMD. E. g. the “Introduction” does not mention nvAMD at all and in “Pathophysiology” the factors VEGF, PEDF, choroidal changes etc. and their implication in nvAMD are not explained (the decrease of PEDF production is only briefly mentioned later in line 377). Currently, the manuscript is more a review about dry AMD or AMD in general.

Consequently, the manuscript has to be revised carefully to a) clearly differentiate between the two forms and their etiology, pathology and treatment, b) unify and correct the use of the terms “AMD”, “nAMD” and “nvAMD” (sometimes neovascular AMD is abbreviated by nAMD sometimes by nvAMD) in text, tables and figures, c) add missing information about nvAMD in the chapters “Introduction”, “Etiology” and “Pathophysiology” and d) add missing information about dry AMD in the chapter “Treatment”. Finally, “neovascular” should be deleted from the title.

OR: The parts describing etiology, pathophysiology etc. about dry AMD should be shortened and information about nvAMD added as aforementioned.

  1. The different prevalence of AMD in different ethnic groups

At several points in the manuscript it is mentioned that “light-skinned” ethnic groups show a higher prevalence of AMD and why. However, I don’t know if there is a kind of official definition for “light-skinned” ethnic groups, but several Asian populations like Chinese or Japanese have a very light skin but lowest prevalence, aren’t they?! In detail:

Lines 47/48 – the term “light-skinned” is used. It is ambiguous in my eyes and the use of “Caucasian” would be less ambiguous.

Line 85 – it is written that dark skin ethnic groups have dark irises and thus, a lower risk to develop AMD. This conclusion is also not true for Asian people with white skin but dark irises.

Lines 83-91 – Consequently, I have doubts about the hypothesis explained here. At least the incoherence regarding light-skinned Asians should be mentioned and it should be more emphasized that this is only a hypothesis.

  1. Genetic risk factors

The etiology misses a paragraph about genetic risk factors (52 known SNPs in 34 loci, CFH, ARMS2/HTRA1, TNF-α polymorphisms, …). However, the appearance of distinct mutations can influence the risk to develop an AMD up to 71%; so, this factor must be detailed.

  1. Current treatment

Lines 321-323 - When current anti-VEGF drugs on the market are mentioned, Beovu (brolucizumab) is missing and consequently, it is also not part of the following discussion about efficacy and duration of the anti-angiogenic effect, though Beovu has been especially developed to reduce the number of annual injections needed.

Table 5 – also here, Beovu is missing.

Line 342 – Conbercept is mentioned “between” the other anti-VEGF drugs on the market. The same applies for Farcizumab a few lines later. It should be clearly differentiated between drugs already on the market and drugs in the experimental phase.

Figure 4 – it is shown that anti-VEGF drugs act on “downstream pathways”, “vascular permeability”, “angiogenic growth factors” and “proinflammatory cytokines”. This is too vague and should be detailed in the figure and, especially, explained in a distinct paragraph.

  1. Gene therapy trials

The list (table 6) misses two nvAMD follow-up clinical trials (NCT03999801and NCT01678872) and (if the authors decide to not focus on nvAMD anymore) one dry AMD trial (NCT03144999).

Minor comments

Abstract: Genetic predisposition should be already mentioned here as risk factor.

Keywords: Would it be not worth to add additional key words that specify the content of the review?

Introduction:

Line 29 – please add a neovascular form

Line 31 – “Wet” should be corrected to “wet”

Line 37 – the abbreviation BrM is used but has not been introduced before, please add this.

Line 47 – it is written that the risk for AMD significantly increases at an age of 75. There is of course no absolute cut-off, but 75 seems a bit late. I would suggest 60-65.

Etiology:

Line 70 – here, hypertension is mentioned as risk factor, though the paragraph is focused on intrinsic factors. Hypertension is more a modifiable factor, though it is also age-related.

Table 3 – the last column is titled “Method of Action”; I would prefer “suggested Method of Action”.

Line 129 – the authors refer to table 4, but the diet is already mentioned in table 3.

Treatment:

Line 325 – the authors write “Anti-VEGF therapy is the mainstay of wet AMD treatment as it affects multiple key pathogenic pathways”. However, in the chapters before, only multiple other pathways were explained but no VEGF-pathway. Thus, the VEGF-pathways has to be detailed (as also written in comment 4.). Additionally, I suggest are more careful wording since the anti-VEGF drugs primarily “only” block the VEGF receptors and do not directly affect downstream pathways.

Conclusion:

Line 388 – The authors define nvAMD as a “rare” subtype; however, ~20% of AMD patients suffer from the neovascular form and this is not a rare event.

Round 2

Reviewer 2 Report

Dear authors, 

the revised version of the manuscript answers very well to my comments and improved significantly. 

Thus, I have no further major comments and only a minor spell check is required.